# Bearing Remaining Useful Life Prediction Based on Naive Bayes and Weibull Distributions

**DOI:** 10.3390/e20120944

**Published:** 2018-12-08

**Authors:** Nannan Zhang, Lifeng Wu, Zhonghua Wang, Yong Guan

**Affiliations:** 1College of Information Engineering, Capital Normal University, Beijing 100048, China; 2Beijing Key Laboratory of Electronic System Reliability Technology, Capital Normal University, Beijing 100048, China; 3Beijing Key Laboratory of Light Industrial Robot and Safety Verification, Capital Normal University, Beijing 100048, China; 4Beijing Advanced Innovation Center for Imaging Technology, Capital Normal University, Beijing 100048, China

**Keywords:** Naive Bayes, remaining useful life, root mean square

## Abstract

Bearing plays an important role in mechanical equipment, and its remaining useful life (RUL) prediction is an important research topic of mechanical equipment. To accurately predict the RUL of bearing, this paper proposes a data-driven RUL prediction method. First, the statistical method is used to extract the features of the signal, and the root mean square (RMS) is regarded as the main performance degradation index. Second, the correlation coefficient is used to select the statistical characteristics that have high correlation with the RMS. Then, In order to avoid the fluctuation of the statistical feature, the improved Weibull distributions (WD) algorithm is used to fit the fluctuation feature of bearing at different recession stages, which is used as input of Naive Bayes (NB) training stage. During the testing stage, the true fluctuation feature of the bearings are used as the input of NB. After the NB testing, five classes are obtained: health states and four states for bearing degradation. Finally, the exponential smoothing algorithm is used to smooth the five classes, and to predict the RUL of bearing. The experimental results show that the proposed method is effective for RUL prediction of bearing.

## 1. Introduction

As the key equipment in the production of products, rotating machinery covers many fields, such as agriculture, machinery manufacturing, industry, electric power, aerospace industry and so on, and plays an important role in the process of industrial production. The emergence of rotating machinery improves production efficiency and reduces energy consumption. However, in the actual production process, due to long-term work and improper operation of parts, mechanical equipment is prone to failure and causes unnecessary losses. The rolling bearing, which plays an indispensable role in the healthy operation of rotating machinery, is an important part of mechanical equipment. The malfunction of rotating machinery is mainly caused by the fault of rolling bearing, and its health state determines the running state of the equipment [1,2,3]. Therefore, the detection of bearing status and the evaluation of life expectancy are very important. Recently, Prognostics and Health Management (PHM) is a promising research direction that can improve the safety and performance of mechanical equipment. PHM predicts the life of the equipment based on actual performance analysis of the equipment. Maintenance of equipment before predicted life can greatly improve the reliability and safety of equipment and reduce the maintenance cost of complex systems. PHM mainly involves mechanical fault diagnosis and residual life prediction. The related fault diagnoses are introduced in the literature [4,5,6]. By predicting the RUL of bearing, the failure of bearing can be found in time. Maintenance and replacement of equipment can improve the operation reliability of mechanical equipment, and it also avoids the loss and casuality caused by bearing failure. In practice, the RUL of bearings is difficult to obtain by experience. It is very important to establish a suitable prediction model for bearing RUL prediction. At present, there are two main methods to predict the RUL of bearings: data-driven prediction and model-based RUL prediction [7]. Model-based and data-driven based methods are widely used for RUL [8].

Among them, the model-based approach predicts the behavior of the system by establishing its internal structure and functions. The model-based method is based on the law of physics, which mainly to analyzes and studies the characteristics of the recessionary components, so as to predict decline trend and the RUL [9]. In the early reliability evaluation method, Lacalle [10] proposed that the error between the theoretical value and the actual value of the system is taken as the correction factor of the fault probability to measure and update the life prediction of the system. In [11], the failure rate model is used to predict reliability. At present, this method is widely used in medical, mechanical and other fields. In [12], considering the uncertainty of structural modeling, a modeling method combining Bayesian and probabilistic structure is proposed to improve the robustness of the system. Liao et al. [13] proposed to combine the proportional hazards model and logistic regression model with bearing RUL prediction. Wahyu et al. [14] proposed degradation parameters or deviation parameters as the object of machine prediction to predict the failure time of a single bearing. The validity and rationality of the degradation model are verified by experiments. The model-based method is developed according to the physical characteristics of the system. When the model is built properly, it can accurately predict the real-time life of the bearing. The model-based approach has achieved some results. However, with the increasing complexity of the system, it becomes more and more difficult to construct the failure model.

The data-driven method does not need to construct complex model, but mainly analyzes the data signal to predict the remaining life of bearing. Because of its simple deployment, data-driven method is widely used in current research. The data-driven method mainly analyzes the data signal to predict the remaining usefulness of the bearing. Analysis of signals vibration is widely used because it can reflect the internal state of degraded bearings and failed bearings [15]. Currently, vibration analysis of bearing RUL mainly includes time domain analysis [16,17,18,19], frequency domain analysis [20] and time-frequency domain analysis [21]. In [22], the degradation trend of bearings is modeled by statistical feature, RMS . Due to the sensitivity to bearing’s degradation information, RMS is regarded as an important degradation index to evaluate the RUL of bearings. However, a single degradation index cannot master the internal performance of the bearing in different periods. Therefore, in this paper, RMS is selected as the main performance degradation index. Through correlation analysis, statistical time domain characteristics with high RMS correlation coefficient is selected as the degradation index. Next, an appropriate prediction model needs to be established. Sun et al. [16] use particle swarm optimisization to optimize the parameters of the support vector machine (SVM) and then predict the remaining life of bearing. Due to single variable SVM’s simple structure and insufficient information, this method often leads to inaccuracy when predicting the result of bearing RUL. Chen et al. [23] proposed a multivariable support vector machine (MSVM) for bearing RUL prediction. He [24] proposes a new method using empirical mode decomposition (EEMD), correlation coefficient analysis, and support vector machine (SVM) to fuse multi-sensor information for bearing fault diagnosis. This method is mainly used when the amount of sample data is small. Although SVM does well in predicting bearing’s RUL, therandomness and complexity of parameters selection is not an easy problem. In recent years, with the successful application of neural network in various fields, Abd Kadir Mahamad et al. [25], proposed artificial neural network (ANN) for bearing RUL prediction. Ben Ali et al. [26], proposes a bearing RUL prediction method based on combining Weibull distribution with ANN. However, due to the uncertainty about the number of hidden layers in the neural network, it is difficult to determine the number of layers in constructing the network. The network keeps trying during training, which leads to randomness of training results. In order to avoid the influence of uncertain number of layers on the training results, Huang Guangbin proposed a new learning algorithm called extreme learning machine (ELM) [27]. ELM is a single hidden layer neural network, which is widely used, by virtue of its simple structure and fast training speed. Fang Liu et al. [17] who proposed a two-layer joint approximate diagonalization of eigen matrices (JADE), which can be regarded as a new degradation index from which redundant features have been eliminated. Then extracted degradation index is passed to the ELM to predict bearing RUL. Next, Fang liu et al. [21] proposed joint phase space reconstruction with JADE to jointly extract sensitive features, and then ELM is used to predict RUL of bearing. ELM greatly shortens the training time, but the randomness in the choice of parameters still caused the randomness of training results. In the current study, Lei Ren [28] proposes a method to compress and calculate the features by using the depth self-encoder, and then uses the depth learning framework to predict the real time life of the bearing. Furthermore, the result of the experiment is achieving better efficiency in bearing RUL prediction. The literature [29] proposes real-time prediction using multi-layer perceptron (MLP) and radial basis functions (RBF). The results show that RBF is superior to MLP in experimental accuracy and time, and results in interesting results. Andres Bustillo et al. [30] proposed to use the popular various Artificial Intelligence (AI) techniques processing sample data set to judge the machine residual life under actual industrial conditions. The experimental results show that the AI technology provides a higher precision to predict the residual life. However, the existing data-driven residual life prediction method does not accumulate knowledge to determine the bearing state. Health status determination is based on expert experience [31]. Bayes is a datahl-driven method based on prior knowledge, which effectively avoids the randomness of results. Naci. Z Gebraeel et al. [32] proposed a Bayesian updating method to update the random parameters of the bearing degradation model and then to develop RUL of degraded device. The method proposed in literature [9] is based on parameters and models. The selection of parameters and the construction of models are very complicated. F. D. Maio’s method et al. [33] applied NB to bearing fault prediction which is a non-parametric data-driven method. When the signal fluctuation of the bearing is large and the accurate classification of bearing cannot be provided, the RUL of the bearing cannot be predicted accurately. According to Reference [34], bearing degradation is rising over time. The running process of bearing is generally divided into three stages: normal operation stage, continuous recession stage, and final failure stage. Therefore, the improved WD is mainly used to fit the bearing signals in different stages to predict the RUL.

This paper mainly considers bearing degradation signal. First, the time domain statistical characteristics are extracted from the vibration signal of the bearing. Then, according to the correlation analysis, the sensitive degradation index of bearing is extracted. Then, the improved WD algorithm is used to fit the degradation index of the fluctuation of bearing in different recession stages which is used as input of NB training stage. The actual degradation data of the bearing is used as test samples. Finally, the results of the time series are smoothed by exponential smoothing, thus the RUL of the bearing is obtained.

The presentation of the paper is as follows. Section 2 briefly describes the correlation analysis, WD and NB. The RUL prediction of bearing, the experimental data and results are presented in Section 3. In Section 4, we draw a conclusion based on the experiment listed in Section 3.

## 2. The Description of the Method

### 2.1. Correlation Analysis

Correlation analysis is a statistical method for analyzing variables [35]. Correlation analysis aims to measure the degree of correlation between variables. The degree of correlation is mainly denoted by the correlation coefficient between variables. The higher the correlation coefficient is, the higher the degree of correlation between the variables is. The correlation coefficients of two stochastic variables *A* and *B* is:(1)$$\rho \left(A,B\right)=\frac{Cov\left(A,B\right)}{\sqrt{D(A)}\sqrt{D(B)}}$$
where Cov(A,B) is the covariance of variables *A* and *B*, D(A) and D(B) are the variance of the variables.

### 2.2. Weibull Distribution

The Weibull Distribution (WD) [36] is widely used in the reliability theory, and the life of most mechanical equipment obeys the WD. WD mainly include two parameters WD and three parameter WD, as shown in Table 1. Where the parameters δ, *k* and *u* are proportional parameters, shape parameters and position parameters, respectively. Where t is the input variable.

An improved WD proposed in [26] is called Universal Failure Rate Function (UFRF), and the UFRF formula is defined as:(2)$$f\left(\delta ,k,b,c\right)=b+c\frac{k}{\delta^{k}}t^{k-1}$$

The parameter δ > 0 is the scale parameter, the *k* > 0 is the shape parameter, the adoption of *c* makes the WD adapt to any range and the adoption of the parameter *b* is to adjust the value of the WD at the beginning.

### 2.3. Naive Bayes

NB is a classification method based on Bayesian rule [37,38]. Bayes and NB are classifiers with prior knowledge. DataSet $X=\{X_{1},X_{2},\ldots ,X_{n}\}$ is known. Where $X_{i}=<A,Y>$, *A* is a property of the dataset $A=\{A_{1},A_{2},\ldots ,A_{t}\}$, *Y* is the category of dataset $Y=\{Y_{1},Y_{2},\ldots ,Y_{m}\}$. When there is unknown data $C=\{C_{1},C_{2},\ldots ,C_{m}\}$, the NB classification method is to assign unknown data *C* to the category *Y* with the largest probability value, i.e., calculate $Max(P\left(Y_{i}/C\right))$. The calculation method of $P(Y_{i}/C)$ is obtained by Bayesian method with prior knowledge:(3)$$P\left(Y_{i}/C\right)=\frac{P\left(C/Y_{i}\right)P\left(Y_{i}\right)}{P(C)},1\leq i\leq m$$
(4)$$P\left(C\right)=\sum_{i=1}^{m}P\left(C/Y_{i}\right)P\left(Y_{i}\right),1\leq i\leq m$$

NB is Bayesian method with independent characteristic conditional, so the probability of P(C) for each class is a constant, it only needs to be calculate $P\left(C/Y_{i}\right)P\left(Y_{i}\right)$. According to known prior knowledge, it can be obtained:(5)$$P\left(Y_{i}\right)=n_{i}/n,1\leq i\leq m$$
(6)$$P\left(C/Y_{i}\right)P\left(Y_{i}\right))=P\left(Y_{i}\right)\prod_{j=1}^{t}P\left(A_{cj}/Y_{i}\right),1\leq i\leq m$$
where $n_{i}$ is the number of class *i* of data set *Y*, and *n* is the total number of data set *Y*. Attribute $A_{c}$ is a continuous property. Usually, the continuous property obeys Gauss distribution $A_{c}∼N\left(u_{c},\delta_{c}^{2}\right)$, Therefore, the Equation for $P(A_{cj}/Y_{i})$ is: (7)$$P\left(A_{cj}/Y_{i}\right)=\frac{1}{\sqrt{2\pi }\delta_{ji}^{}}exp\left\{-\frac{{\left(A_{cj}-u_{ij}\right)}^{2}}{2\delta_{ji}^{2}}\right\},1\leq i\leq m$$
where $u_{c}$ and $\delta_{c}^{2}$ are the mean and variance of the dataset *X*, respectively.
(8)$$u_{ij}=\frac{\sum_{g=1}^{n_{i}}X_{gj}^{\left(i\right)}}{n_{i}}$$
(9)$$\delta_{ij}^{2}=\frac{\sum_{g=1}^{n_{i}}{\left(X_{gj}^{\left(i\right)}-u_{i}j\right)}^{2}}{n_{i}-1}$$

The discriminant function of the NB can be obtained as follows:(10)$$P\left(Y_{i}/C\right)=\frac{P\left(Y_{i}\right)\prod_{j=1}^{t}\frac{1}{\sqrt{2\pi }\delta_{ji}}exp\left\{-\frac{{\left(A_{cj}-u_{ij}\right)}^{2}}{2\delta_{ji}^{2}}\right\}}{\sum_{i=1}^{m}P\left(Y_{i}\right)\prod_{j=1}^{t}\frac{1}{\sqrt{2\pi }\delta_{ji}}exp\left\{-\frac{{\left(A_{cj}-u_{ij}\right)}^{2}}{2\delta_{ji}^{2}}\right\}}$$

Therefore, the category of unknown data *C* is judged as the category with the maximum value of discriminant function $P(Y_{i}/C)$.

## 3. Prediction of Bearing RUL

In this paper, the prediction model framework of bearing RUL is divided into three stages as shown in Figure 1.

### 3.1. Feature Construction

#### 3.1.1. Signal Acquisition

To verify the effectiveness of our method, the full-cycle bearing data used in this paper is from the Intelligent Maintenance Center of the University of Cincinnati [39]. The full cycle data (run-to-failure) acquisition device of the bearing is shown in Figure 2. One is the real figure of the bearing and the other is the sensor placement illustration of the bearing. In addition, sensor placement illustration and real figure are corresponding.

As it is shown in Figure 2, the bearing experimental test platform consists of four Rexnord ZA-2115 double row test bearings (Bearing 1–4) mounted on a rotating shaft. The shaft is driven by AC motor and the bearing is always kept at a constant speed of 2000 revolutions per minute. Furthermore, radial load system of 2724 kg is loaded on the bearing and the drive shaft. In addition, there are two ICP-based Accelerometers(model: PCB 353B33 high sensitivity quartz ICP) produced by PCB USA to measure the vibration data along the x and y of channel signals. The vibration data were collected by NI DAQ card 6062E and recorded every 10 min using 20 kHz sampling frequency. The degradation of bearing s is mainly determined by the debris collected by magnetic plugs. When a certain amount is reached, the test platform ends the test. After 35 days of testing, a total of 3 sets of run-to-failure data were collected. Each set contains 20,480 sampling points. In this experiment, the bearing run-to-failure data of three data sets were adopted, as shown in Table 2. Figure 3 shows the vibration data of the entire cycle of three datasets from normal to fault.

It can be seen from Figure 3 that the signal frequency of bearing 1, bearing 3 and bearing 4 is low at the beginning and the vibration frequency increases gradually over time. However, the amplitude analysis of the signal cannot get the RUL of the bearing. Furthermore, there is a certain noise in these signals. It is very difficult to analyze these signals directly. So the signal is processed preliminarily.

#### 3.1.2. Feature Extraction

The time domain signal is a waveform signal that changes with time and contains the state information of the bearing in the waveform. The state of bearing can be diagnosed by analyzing time-domain waveform. However, in the real working condition, the bearing is mixed with the noise during the operation, so this paper uses the statistical time domain method to extract sensitive features from signals. The health status degradation information of the bearing is extracted by statistical calculation. As shown in Table 3, there are a total of 16 statistical time-domain features.

The 16 original statistical time-domain features (as shown in Table 1) extracted from the vibration signal are selected, and described in Figure 4, Figure 5 and Figure 6. The time-domain features of $F_{1}-F_{8}$ are RMS, the average value, absolute mean, average power, square amplitude, peak, peak-to-peak and variance, respectively. The time-domain hlfeatures of $F_{9}-F_{16}$ are standard deviation, skewness, kurtosis, waveform, Crest index, impluse index, margin index and skewness index, respectively.

It can be seen from Figure 4, Figure 5 and Figure 6 that not all features of bearing degradation data $F_{1}-F_{16}$ are robust in the process of degradation. For example, $F_{6}$ in Figure 4 and Figure 5 cannot well present the degradation process of bearing, but the features $F_{4}$ forms well is about the degradation process of bearing. So, we need to find the features that can better present the degradation process of the bearing. Therefore, it is necessary to choose a suitable and robust feature for the RUL prediction.

#### 3.1.3. Feature Selection

Before life prediction modeling, the appropriate features are very important for reflecting the bearing degradation process. Therefore, it is necessary to select the characteristics of 16 time-domain features and select the appropriate features to predict the RUL of the bearing. Time domain features of RMS can effectively reflect the overall health status of bearings. Furthermore, with the degradation of the bearing, the degradation trend of RMS gradually increased. So, RMS is often used as the main features for bearing trend analysis and RUL prediction [40,41]. This paper takes RMS as the main feature, and uses correlation coefficients to extract the statistical time domain features consistent with the RMS trend, and then to predict the RUL of bearing.

In this paper, the correlation coefficient is used to extract the high-correlation time domain features with RMS. The flow chart is shown in Figure 7.

First of all, we construct the features matrix $M_{nt}=\left\{M_{1},M_{2},\ldots ,M_{t}\right\}$, where $M_{i}=\left\{f_{1i},f_{2i},\ldots ,f_{ni}\right\}$. *n* is the number of samples and *t* is the number of features attributes. $M_{nt}$ is written as:(11)$$M=\left[\begin{array}{c} f_{11}\;f_{12}\;\ldots \;f_{1t}\\ f_{21}\;f_{22}\;\cdots \;f_{2t}\\ \ldots \;\ldots \;\ldots \;\ldots \\ f_{n1}\;f_{n2}\;\ldots \;f_{nt} \end{array}\right]$$

Then, the correlation coefficient matrix R1t between RMS and Mnt is calculated.
(12)$$R=\left[\begin{array}{c} r_{11}\;r_{12}\;\ldots \;r_{1t} \end{array}\right]$$

Among them, the correlation coefficient *r* can be calculated according to Equation (Equation 1), then $r_{1i}=\rho \left(RMS,F_{i}\right)$. Finally, the threshold *T* is set to determine the relationship between correlation coefficient matrix *R* and *T*. If $r_{1i}>T$, it shows that the feature $F_{i}$ has a strong correlation with RMS. Otherwise, the correlation between *F* and RMS is low, then the *i*th column in the feature matrix is dropped.

In the experiment of this paper, there are 16 feature vectors. Therefore, the correlation coefficients between RMS and 16 feature vectors need to be calculated. In general, the correlation coefficient of the two vectors is greater than 0.9, and we consider that these two vectors are significantly correlated. In this paper, we need to get features of high correlation and RMS. We set the threshold *T* = 0.9 to filter off the features of low correlation, and the dimension of the features is reduced to 6. The six features are $F_{1},\;F_{3},\;F_{4},\;F_{5},\;F_{8}$ and $F_{9}$. Correlation coefficient between bearing features and RMS are as shown in Table 4.

### 3.2. Classification Model Construction

In this paper, NB is used to train and test the bearing degradation data. The training data mainly use UFRF to fit the bearing data of degradation trend. In the testing process, the real bearing degradation data are used for classification and then for predicting the RUL.

The overall degradation of the bearing is increasing over time. In Figure 4, Figure 5 and Figure 6, it can be seen that the extracted six time-domain features curves of $F_{1},\;F_{3},\;F_{4},\;F_{5},\;F_{8}$ and $F_{9}$ have a large number of fluctuations, which make it difficult for the structural classification to predict the RUL of the bearing. There are many reasons for these fluctuations, such as noise, speed and so on. In order to avoid the influence of these fluctuations, the UFRF proposed in literature [26] is used to smooth and fit the features smoothing increases with time.

However, the bearing degradation process can be divided into three stages and each stage is different. The first stage is the normal operation stage, and the curve of RMS features of bearings in this stage has no obvious change, and there is no degeneration. The second stage is the continuous recession stage, where the degradation is relatively obvious, but the degradation is a slow and continuous process. Generally there will be no significant degradation. The third stage is the final failure stage. Bearing degradation is severe and its fluctuation will be especially obvious. It can be seen from above that the degradation process of each degradation stage is different. Therefore, in this paper, the UFRF with different parameters is used to fit thefeatures of the degradation process at different stages. However, the overall degradation is increasing over time, consistent with the bearing degradation process. Reference [42] shows that the starting point of bearing 1 and bearing 3 failure is at the 586th point and 1808th, respectively. Then the decline trend of bearing degradation curve shows down in the following time. Furthermore, then the degradation curve of the bearing will change rapidly after reaching the failure threshold (950th and 2120th). Literature [21] shows that the three-stages time ratio of four is 0.5:0.2:0.3. Table 5, Table 6 and Table 7 show the parameters of different stages of bearing 1 feature extraction. The selection of parameters is first randomly generated according to the distribution rules of the WD distribution, and then the optimal parameters are obtained by adjusting the parameters according to the real data.

Through Table 5, Table 6 and Table 7, features according with bearing degradation is fitted. In Figure 8, Figure 9 and Figure 10, the selected features are consistent with the bearing degradation process. After fitting, the fitted features are constructed for the following training. NB is a supervised learning method, so it is necessary to label the training data. Figure 11 shows us adding labels of varying degrees of degradation according to the RMS degradation process. As can be seen from the Figure 11a, there are six types of labels, among which label 1 represents normal bearing data, label 2, 3, 4 and 5 respectively represent 25%, 50%, 90% and complete failure, respectively. In Figure 11b,c, the six type of labels represent 0, 30%, 60%, 90%, 100% and 0, 40%, 80%, 95%, 100%, respectively. After UFRF fitting, the fitted feature is used as the input of NB. In the test phase, the true feature of bearing degradation is as the input.

Through Table 4, features according with bearing degradation is fitted. In Figure 6, the selected features are consistent with the bearing degradation process. After fitting, the fitted features are constructed for the following training. NB is a supervised learning method, so it is necessary to tag the training data. Figure 11 shows us adding labels of varying degrees of degradation according to the RMS degradation process. As can be seen from the Figure 11a, there are six types of labels, among which label 1 represents normal bearing data, label 2, 3, 4 and 5 respectively represent 25%, 50%, 90% and complete failure, respectively. In Figure 11b,c, the six type of labels represent 0, 30%, 60%, 90%, 100% and 0, 40%, 80%, 95%, 100%, respectively. After UFRF fitting, the fitted feature is used as the input of NB. In the test phase, the true feature of bearing degradation is as the input.

Then, according to the classification concept of NB, the bearing data is divided into five categories (label 1–5). The descriptions of bearing data are shown in Table 8, Table 9 and Table 10.

After fitting, the training data set was obtained, and then the NB classification model was trained. The real bearing data is used as the model input, and the test results are obtained, as shown in Table 11, Table 12 and Table 13. Table 14 shows the comparison results of bearing 1. It can be seen from the table that the result of NB classification is better than reference [26].

#### RUL Construction

After the classification, we have obtained the category. However, the results cannot predict the RUL of bearings directly. Therefore, a new approach is needed to convert the category results of the time series into the degradation trend of the time series, and then to predict the RUL of the bearing.

The exponential smoothing is proposed by Brown, who believes that the time series is stable and regular [43]. So the time series can be reasonably forecasted and exponential smoothing is a common method in prediction. Therefore, this paper proposes that exponential smoothing method is used to obtain the real degradation process of bearings, so as to predict the RUL of bearings. The equation is as follows:(13)$$s_{t}=zl_{t}+\left(1-z\right)s_{t-1}$$
where *z* is the smoothing constant and $s_{t}$ the smoothing value of time *t*, is the real value of time *t*. $s_{t-1}$ the smoothing value of time t−1. The smoothing constant *z* is very important for the smoothing level, which determines the gap response speed between the predicted value and the actual result.

The range of the smoothness constant *z* is from 0 to 1. In general, the closer the smoothing constant is to 0, the stronger the smoothing effect is. The smoothing constant in this paper is a value close to 0. The experimental results show that when the smoothing constants *z* of bearings 1, 3 and 4 are 0.08, 0.05 and 0.07 respectively, RUL prediction achieve best in this paper, as shown in Figure 12. Figure 12a–c are the predicted results of bearing 1, 3 and 4, respectively. As can be seen from the Figure 12, after smoothing, we can get the RUL of each point.

Error is used as a measurement index to predict bearing precision, and its equation is:(14)$$e=\frac{1}{n}\sum_{t=1}^{n}\left|s_{t}-l_{t}\right|$$
where $s_{t}$ is the predicted RUL, and $l_{t}$ is the real RUL of the bearing.

Table 15 is a comparison of the experimental results of the bearing degradation data set. It can be concluded from the experimental results in the table that the algorithm proposed in this paper is more accurate in predicting the RUL of the bearing.

## 4. Conclusions

RUL prediction is very important in improving the safety, reliability, availability and maintainability of rotating machinery. Through the prediction of residual life, the possible faults can be detected and predicted in time, and the rotating machinery can be repaired so as to prolong the service life of rotation. This paper starts with bearing vibration signal, combined with data drive technology and fault prediction method to predict the residual life of bearing. Firstly, feature extraction and selection of vibration signal are performed. Then, according to the law of bearing degradation, the bearing data are divided into three stages, and the bearing data are divided into different categories. Next, the bearing degradation data were classified using NB. Finally, an exponential distribution is used to predict bearing’s RUL. In order to verify the effectiveness of the experiment, real bearing degradation data were used to perform the experiment. The experimental results show that the NB is effective in predicting the RUL of the bearing. In this paper, the residual life of bearings is predicted based on the classification method, and it is hoped that the remaining life of the bearing can be predicted directly in the future.

## Figures and Tables

**Figure 1 entropy-20-00944-f001:**
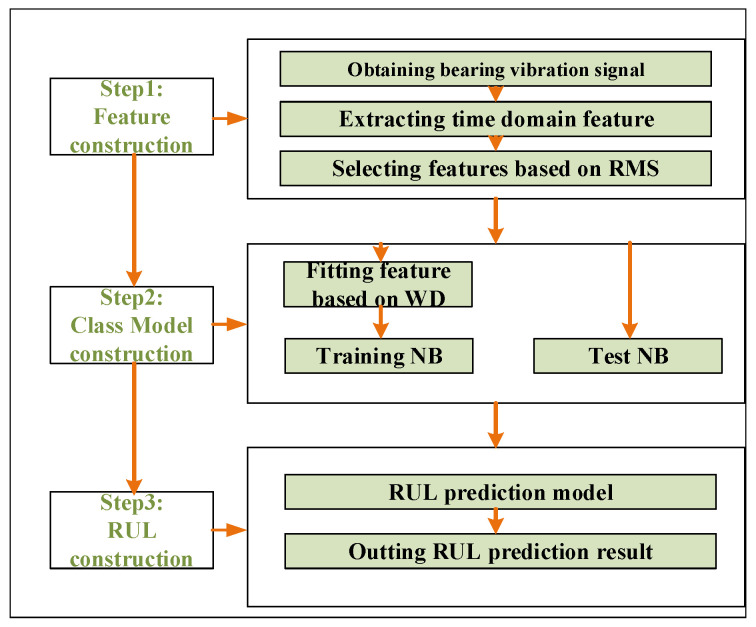
Bearing remaining useful life (RUL) prediction model framework.

**Figure 2 entropy-20-00944-f002:**
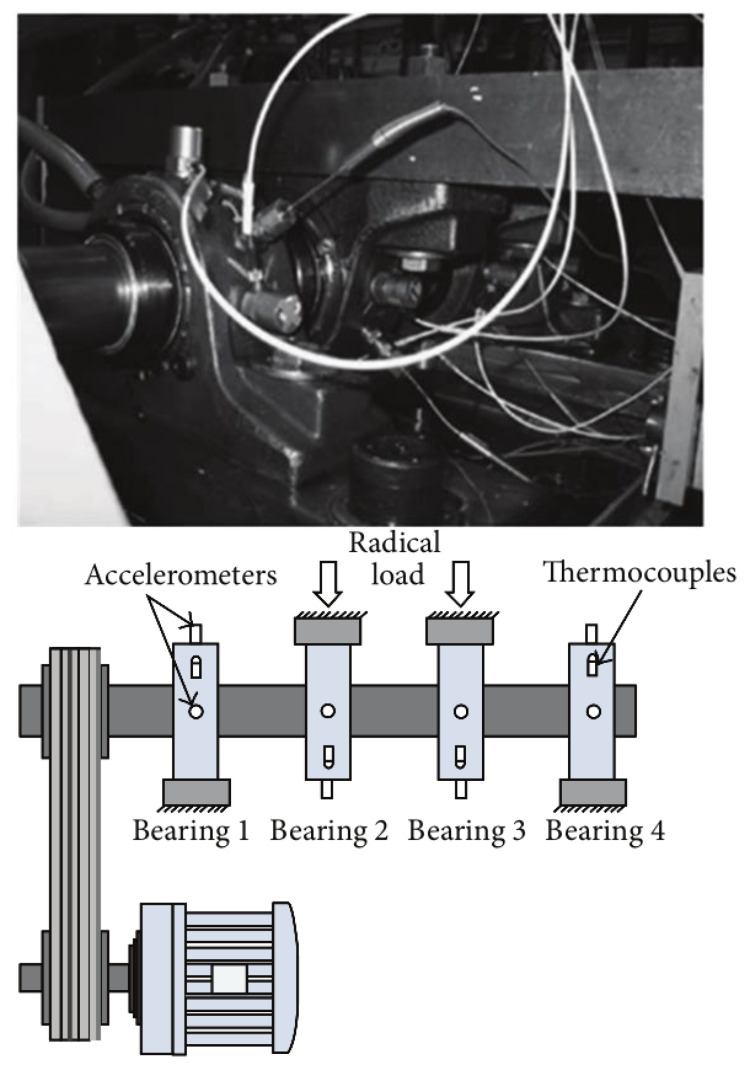
The bearing experimental device and sensor placement illustration [39].

**Figure 3 entropy-20-00944-f003:**
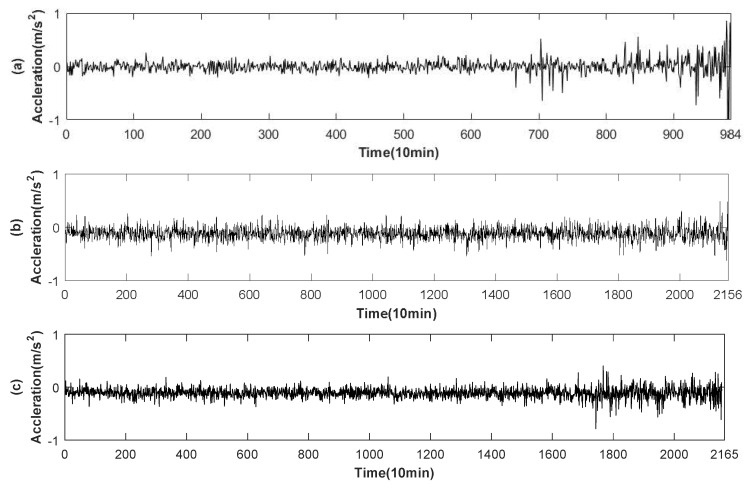
Bearing run-to-failure data. (**a**) Bearing 1 of Set No. 2 is vibration signals ending with an outer race failure; (**b**) Bearing 3 of Set No. 1 is vibration signals ending with an inner race defect; (**c**) Bearing 4 of Set No. 1 is vibration signals ending with roller element defect.

**Figure 4 entropy-20-00944-f004:**
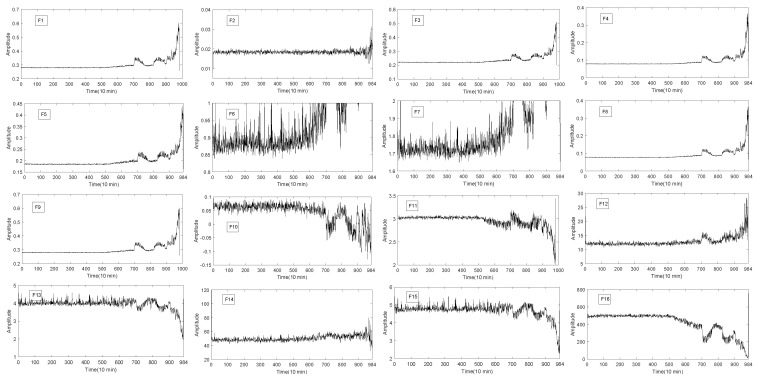
The 16 time-domain features of bearing 1.

**Figure 5 entropy-20-00944-f005:**
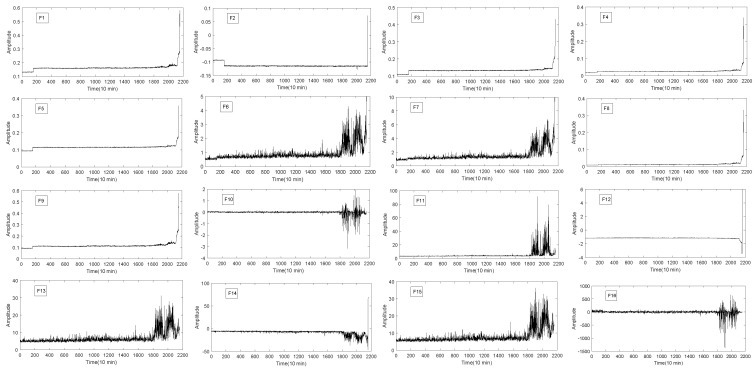
The 16 time-domain features of bearing 3.

**Figure 6 entropy-20-00944-f006:**
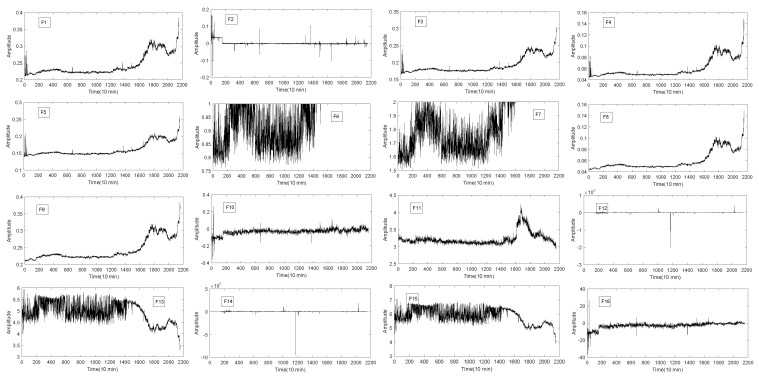
The 16 time-domain features of bearing 4.

**Figure 7 entropy-20-00944-f007:**
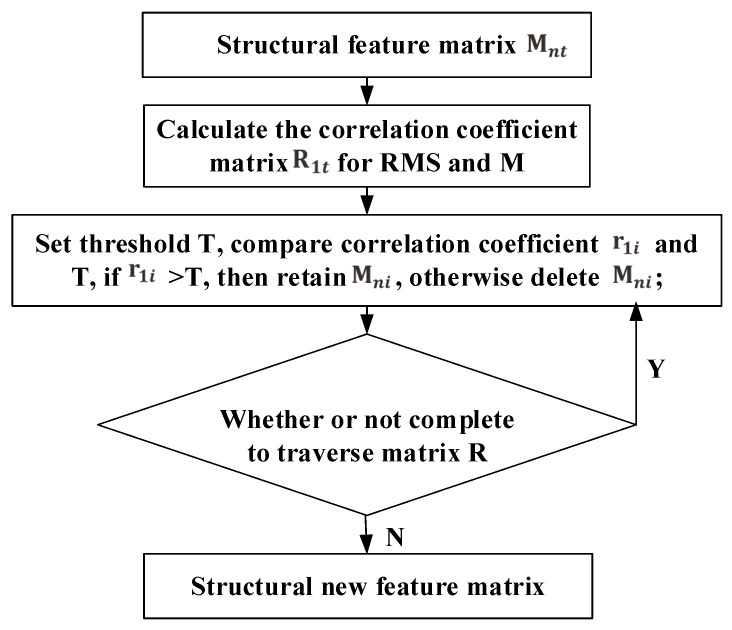
Based on root mean square (RMS) feature selection flow chart.

**Figure 8 entropy-20-00944-f008:**
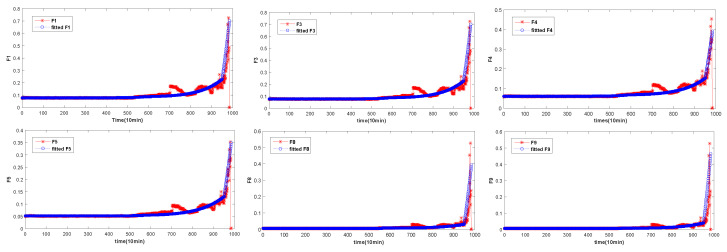
Fitted feature of bearing 1.

**Figure 9 entropy-20-00944-f009:**
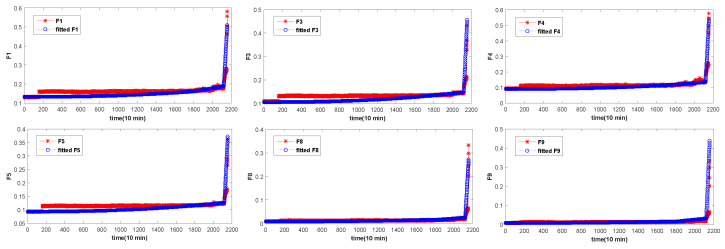
Fitted feature of bearing 3.

**Figure 10 entropy-20-00944-f010:**
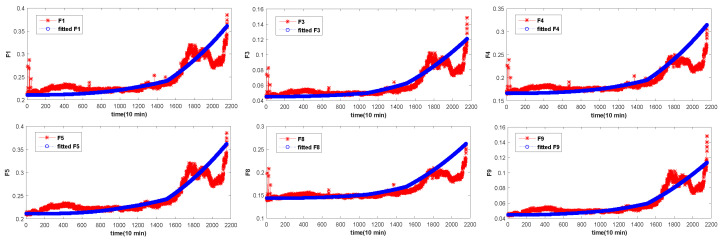
Fitted feature of bearing 4.

**Figure 11 entropy-20-00944-f011:**
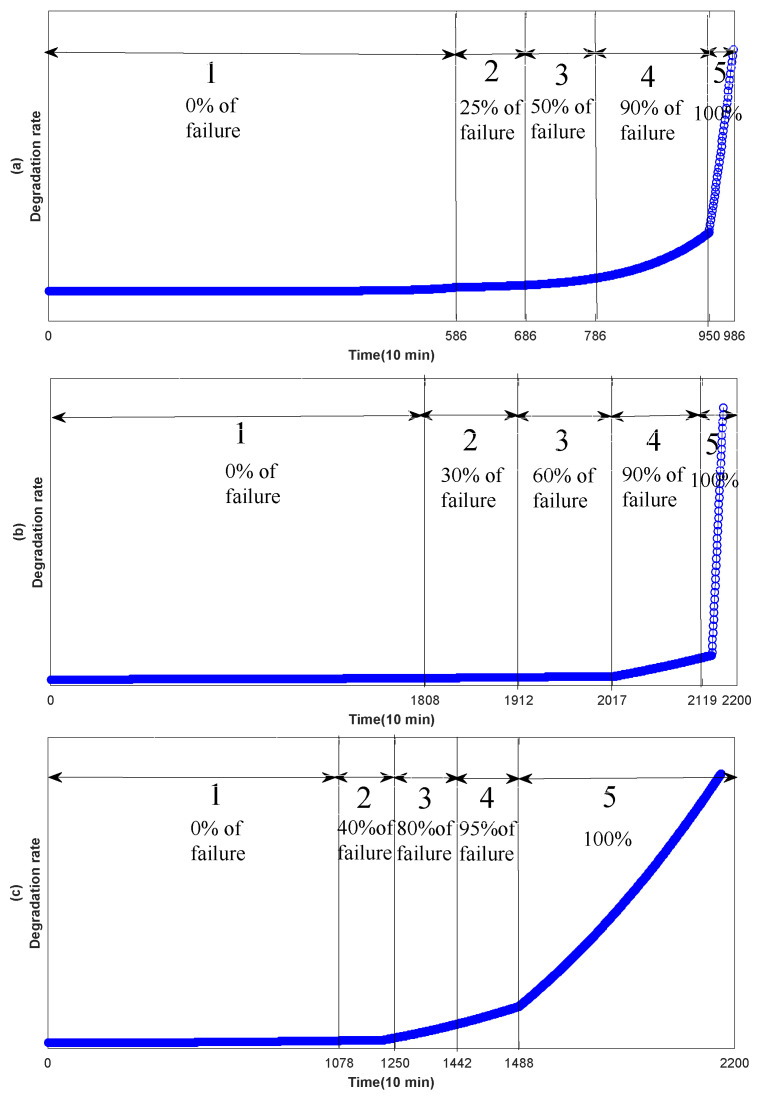
Classification of degradation data for bearing. (**a**) The class label for bearing 1; (**b**) The class label for bearing 3; (**c**) The class label for bearing 4.

**Figure 12 entropy-20-00944-f012:**
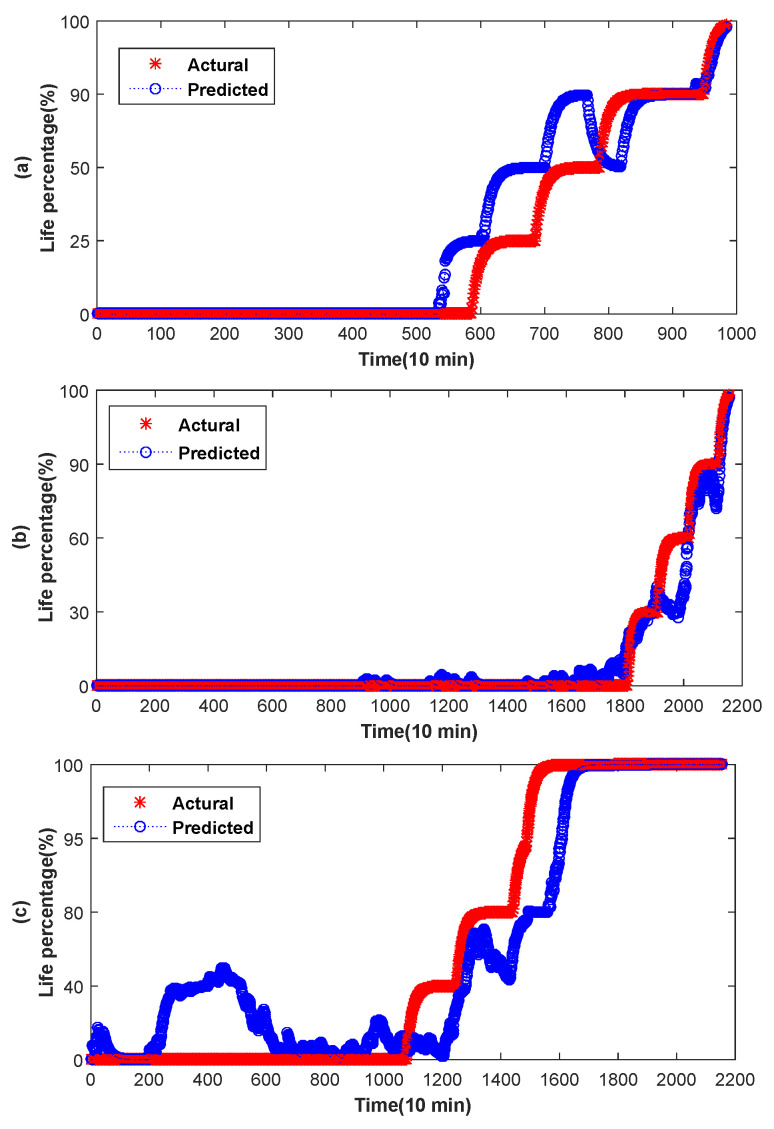
Fitted feature of bearing 3.

**Table 1 entropy-20-00944-t001:** Weibull distribution.

Two-Parameter Weibull	Three-Parameter Weibull
$f\left(\delta ,k\right)=\frac{k}{\delta }{\left(\frac{t}{\delta }\right)}^{k-1}e^{-{\left(\frac{t}{\delta }\right)}^{k}}$	$f\left(\delta ,k\right)=\frac{k}{\delta }{\left(\frac{t-u}{\delta }\right)}^{k-1}e^{-{\left(\frac{t-u}{\delta }\right)}^{k}}$

**Table 2 entropy-20-00944-t002:** Description of bearing run-to-failure data set.

Bearing Data	Speed (rpm)	Number of Samples	Type of Fault
Set No. 2 Bearing 1	2000	944	Out race
Set No. 1 Bearing 3	2000	2156	Inner race
Set No. 1 Bearing 4	2000	2156	Roller

**Table 3 entropy-20-00944-t003:** Time domain analysis of bearing run-to-failure data.

Number	Characteristic Equation	Number	Characteristic Equation
1	$F_{1}=\sqrt{\frac{\sum_{n=1}^{d}{\left(X\left(n\right)\right)}^{2}}{d}}$	2	$F_{2}=\frac{\sum_{n=1}^{d}X\left(n\right)}{d}$
3	$F_{3}=\frac{\sum_{n=1}^{d}\left|X\left(n\right)\right|}{d}$	4	$F_{4}=\frac{\sum_{n=1}^{d}{\left(X\left(n\right)\right)}^{2}}{d}$
5	$F_{5}={\left(\frac{\sum_{n=1}^{d}\sqrt{\left|X\left(n\right)\right|}}{d}\right)}^{2}$	6	$F_{6}=max\left|X(n)\right|$
7	$F_{7}=max\left(X\left(n\right)\right)-min\left(X\left(n\right)\right)$	8	$F_{8}=\frac{\sum_{n=1}^{d}{\left(X\left(n\right)-F_{1}\right)}^{2}}{d-1}$
9	$F_{9}=\sqrt{\frac{\sum_{n=1}^{d}{\left(X\left(n\right)-T_{1}\right)}^{2}}{d-1}}$	10	$F_{10}=\frac{\sum_{n=1}^{d}{\left(X\left(n\right)\right)}^{3}}{d}$
11	$F_{11}=\frac{\sum_{n=1}^{d}{\left(X\left(n\right)\right)}^{4}}{d}$	12	$F_{12}=\frac{F_{3}}{F_{2}}$
13	$F_{13}=\frac{F_{6}}{F_{3}}$	14	$F_{14}=\frac{F_{6}}{F_{2}}$
15	$F_{15}=\frac{F_{6}}{F_{5}}$	16	$F_{16}=\frac{F_{10}}{F_{9}^{3}}$

**Table 4 entropy-20-00944-t004:** Correlation coefficient between bearing features and RMS.

	$F_{1}$	$F_{3}$	$F_{4}$	$F_{5}$	$F_{8}$	$F_{9}$
Set No. 2 Bearing 1	1	0.9989	0.9910	0.9973	0.9910	1
Set No. 1 Bearing 3	1	0.9985	0.9892	0.9962	0.9937	0.9992
Set No. 1 Bearing 4	1	0.9983	0.9987	0.9962	0.9955	0.9959

**Table 5 entropy-20-00944-t005:** Universal Failure Rate Function (UFRF) parameters of bearing 1 in different degradation stages.

Feature	Stage	δ	*k*	*b*	*c*
	Normal stage	310.0209	12.2517	0.0782	$1.48\times 10^{-4}$
$F_{1}$	Continuous recession stage	310.0209	10.1917	0.1555	$1.48\times 10^{-4}$
	Final failure stage	289.0209	11.2017	−0.8362	$1.48\times 10^{-4}$
	Normal stage	310.0209	12.2517	0.0782	$1.48\times 10^{-4}$
$F_{3}$	Continuous recession stage	310.0209	10.1917	0.1555	$1.48\times 10^{-4}$
	Final failure stage	289.0209	11.2017	−0.8362	$1.48\times 10^{-4}$
	Normal stage	297.0209	11.5217	0.0616	$1.38\times 10^{-4}$
$F_{4}$	Continuous recession stage	310.0209	9.8917	0.0663	$1.28\times 10^{-4}$
	Final failure stage	270.0209	10.8017	−0.4843	$1.38\times 10^{-4}$
	Normal stage	310.0209	10.9817	0.0517	$1.48\times 10^{-4}$
$F_{5}$	Continuous recession stage	295.0209	9.3017	0.2335	$1.48\times 10^{-4}$
	Final failure stage	296.0209	10.8517	−0.3988	$1.48\times 10^{-4}$
	Normal stage	310.0209	10.06	0.0061	$1.48\times 10^{-4}$
$F_{8}$	Continuous recession stage	295.0209	10.3017	0.0075	$1.28\times 10^{-5}$
	Final failure stage	291.0209	11.1317	−0.8164	$1.38\times 10^{-4}$
	Normal stage	310.0209	12.2517	0.0782	$1.48\times 10^{-4}$
$F_{9}$	Continuous recession stage	295.0209	10.5517	0.00755	$1.38\times 10^{-5}$
	Final failure stage	288.0209	11.1617	−0.7486	$1.38\times 10^{-4}$

**Table 6 entropy-20-00944-t006:** UFRF parameters of bearing 3 in different degradation stages.

Feature	Stage	δ	*k*	*b*	*c*
	Normal stage	78	3.4	0.023	$3.5\times 10^{-4}$
$F_{1}$	Continuous recession stage	97	3.7917	0.0426	$3.5\times 10^{-4}$
	Final failure stage	100	4.7017	4.9300	$0.13\times 10^{-2}$
	Normal stage	78	3.4	0.018	$3.55\times 10^{-4}$
$F_{3}$	Continuous recession stage	97	3.6417	0.0196	$3.55\times 10^{-4}$
	Final failure stage	95	4.7017	6.2923	$0.13\times 10^{-2}$
	Normal stage	78	3.4	0.018	$3.55\times 10^{-4}$
$F_{4}$	Continuous recession stage	97	0.0196	0.0196	$3.55\times 10^{-4}$
	Final failure stage	100	4.7017	6.2923	$0.13\times 10^{-2}$
	Normal stage	78	3.33	0.018	$3.55\times 10^{-4}$
$F_{5}$	Continuous recession stage	100	3.5417	0.0148	$3.55\times 10^{-4}$
	Final failure stage	78	4.6017	3.9144	$0.13\times 10^{-2}$
	Normal stage	78	3.17	0.009	$3.55\times 10^{-4}$
$F_{8}$	Continuous recession stage	100	3.5217	0.0148	$3.55\times 10^{-4}$
	Final failure stage	78	4.6017	0.1864	$0.13\times 10^{-2}$
	Normal stage	78	3.03	0.004	$3.55\times 10^{-4}$
$F_{9}$	Continuous recession stage	97	3.5417	0.0196	$3.55\times 10^{-4}$
	Final failure stage	95	4.7017	6.2923	$0.13\times 10^{-2}$

**Table 7 entropy-20-00944-t007:** UFRF parameters of bearing 4 in different degradation stages.

Feature	Stage	δ	*k*	*b*	*c*
	Normal stage	96	3.6	0.2138	$3.55\times 10^{-4}$
$F_{1}$	Continuous recession stage	91	3.417	0.2218	$3.5\times 10^{-4}$
	Final failure stage	97	3.6317	0.169	$0.13\times 10^{-2}$
	Normal stage	96	3.5	0.0449	$3.55\times 10^{-4}$
$F_{3}$	Continuous recession stage	92	3.537	0.0425	$3.55\times 10^{-4}$
	Final failure stage	95	3.5317	0.0073	$0.13\times 10^{-2}$
	Normal stage	96	3.5	0.0449	$3.55\times 10^{-4}$
$F_{4}$	Continuous recession stage	92	3.537	0.0425	$3.55\times 10^{-4}$
	Final failure stage	95	3.5317	0.0073	$0.13\times 10^{-2}$
	Normal stage	96	3.73	0.2098	$3.55\times 10^{-4}$
$F_{5}$	Continuous recession stage	93	3.5417	0.213	$3.55\times 10^{-4}$
	Final failure stage	95	3.7317	0.1352	$0.13\times 10^{-2}$
	Normal stage	94	3.53	0.1423	$3.55\times 10^{-4}$
$F_{8}$	Continuous recession stage	93	3.747	0.133	$3.55\times 10^{-4}$
	Final failure stage	95	3.5317	0.1109	$0.13\times 10^{-2}$
	Normal stage	95	3	0.0494	$3.55\times 10^{-4}$
$F_{9}$	Continuous recession stage	91	3.317	0.0469	$3.55\times 10^{-4}$
	Final failure stage	92	3.3817	0.0192	$0.13\times 10^{-2}$

**Table 8 entropy-20-00944-t008:** Description of bearing 1 data set.

Data Type	The Number of Training	The Number of Testing	Lable
Normal	585	585	1
25% of failure	100	100	2
50% of failure	100	100	3
90% of failure	164	164	4
100% of failure	35	35	5
Total	984	984	5

**Table 9 entropy-20-00944-t009:** Description of bearing 3 data set.

Data Type	The Number of Training	The Number of Testing	Lable
Normal	1808	1808	1
30% of failure	104	104	2
60% of failure	105	105	3
90% of failure	102	102	4
100% of failure	37	37	5
Total	2156	2156	5

**Table 10 entropy-20-00944-t010:** Description of bearing 4 data set.

Data Type	The Number of Training	The Number of Testing	Lable
Normal	1077	1077	1
40% of failure	172	172	2
80% of failure	192	192	3
95% of failure	46	46	4
100% of failure	668	668	5
Total	2156	2156	5

**Table 11 entropy-20-00944-t011:** Class classification accuracy of bearing 1.

Class	1	2	3	4	5	Total
Number of class	585	100	100	164	35	984
Well number of class	531	100	20	117	28	796
Accuracy	91.1%	100%	20%	71.3%	80%	80.9%

**Table 12 entropy-20-00944-t012:** Class classification accuracy of bearing 3.

Class	1	2	3	4	5	Total
Number of class	1808	104	105	102	37	2156
Well number of class	1753	84	17	52	37	1943
Accuracy	97%	80.8%	16.2%	51%	100%	90.2%

**Table 13 entropy-20-00944-t013:** Class classification accuracy of bearing 4.

Class	1	2	3	4	5	Total
Number of class	1077	172	192	46	668	2156
Well number of class	669	47	71	0	570	1357
Accuracy	62.2%	2.32%	36.98%	0%	85.33%	62.94%

**Table 14 entropy-20-00944-t014:** Comparison of bearing test results.

Set No. 2—Bearing 1	Accurancy
NB	80.9%
Reference [26]	74.2%

**Table 15 entropy-20-00944-t015:** Comparison of prediction error by different methods.

Data	Algorithm	*e*
Set No. 2—Bearing 1	NB+UFRF	0.1173
	Referfece [42]	4.61
Set No. 2—Bearing 3	NB+UFRF	0.0402
	Reference [42]	0.98
Set No. 2—Bearing 4	NB+UFRF	0.6345

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
