# Peer review of "Bearing Remaining Useful Life Prediction Based on Naive Bayes and Weibull Distributions"

_entropy, 2018, doi:10.3390/e20120944_

Round 1

Reviewer 1 Report

1) Figures should have better quality. 

Figure 2 please add arrows to real figure

2) Please add SI units (if any)

3) Why the topic is essential. Please add image of application.

4) What does it mean Time (10min) it is better to write min or seconds

5) The authors should cite new references (2016-2019 Web of Science). 

At least new 25 referneces.

Please show that you have new knowledge.

for example 

about fault diagnosis/naive bayes/bearings etc.

Fault diagnosis of single-phase induction motor based on acoustic signals

By:Glowacz, A (Glowacz, Adam)[ 1 ]

MECHANICAL SYSTEMS AND SIGNAL PROCESSING

Volume: 117  Pages: 65-80

DOI: 10.1016/j.ymssp.2018.07.044

Published:FEB 15 2019

Multi-dimensional variational mode decomposition for bearing-crack detection in wind turbines with large driving-speed variations

By:Li, ZX (Li, Zhixiong)[ 1,3,4 ] ; Jiang, Y (Jiang, Yu)[ 3,4,5 ] ; Guo, Q (Guo, Qiang)[ 6 ] ; Hu, C (Hu, Chao)[ 1,2 ] ; Peng, ZX (Peng, Zhongxiao)[ 5 ]

RENEWABLE ENERGY

Volume: 116  Pages: 55-73  Part: B

DOI: 10.1016/j.renene.2016.12.013

Published:FEB 2018

Optimised ensemble empirical mode decomposition with optimised noise parameters and its application to rolling element bearing fault diagnosis

By:Zhang, C (Zhang, Chao)[ 1,2 ] ; Li, ZX (Li, Zhixiong)[ 2,3,4 ] ; Chen, S (Chen, Shuai)[ 1 ] ; Wang, JG (Wang, Jianguo)[ 1 ] ; Zhang, XG (Zhang, Xiaogang)[ 2 ]

INSIGHT

Volume: 58  Issue: 9  Pages: 494-501

DOI: 10.1784/insi.2016.58.9.494

Published:SEP 2016

about NB

Diagnostics of Rotor Damages of Three-Phase Induction Motors Using Acoustic Signals and SMOFS-20-EXPANDED

By:Glowacz, A (Glowacz, Adam)[ 1 ]

ARCHIVES OF ACOUSTICS

Volume: 41  Issue: 3  Pages: 507-515

Published:2016

about Weibull 

Synthetic fault factor features under Weibull stochastic interference

By:Zhang, YG (Zhang, Yagang)[ 1,2 ] ; Wang, PH (Wang, Penghui)[ 1 ] ; Zhang, X (Zhang, Xue)[ 1 ] ; Wang, C (Wang, Can)[ 1 ]

INTERNATIONAL JOURNAL OF ELECTRICAL POWER & ENERGY SYSTEMS

Volume: 77  Pages: 19-24

DOI: 10.1016/j.ijepes.2015.11.033

Published:MAY 2016

Author Response

Response to the referee’s comments

Thank you very much for your comments about our paper(Manuscript ID: entropy-396966. Title: Bearing remaining useful life prediction based on Naive Bayes and Weibull distributions)submitted to Entropy. We have checked the manuscript and revised it according to the comments. We submit here the revised manuscript as well as a list of changes. If you have any question about this paper, please don’t hesitate to let me know. Your efforts in the review process of this manuscript are greatly appreciated.

Comments.1) Figures should have better quality. Figure 2 please add arrows to real figure

Response: Thank you for your question. We have revised Figure 2, and we have added the question to the paper (Refer to lines 124 to 138 in the paper).

Comments.2) Please add SI units (if any)

Response: Thanks for the good suggestion! The unit used in this paper is SI units.

Comments.3) Why the topic is essential. Please add image of application.

Response: Thanks for your comments. We have added this question to the paper(Refer to lines 16 to 25 in the paper).  

Comments.4) What does it mean Time (10min) it is better to write min or seconds

Response:  Thanks for your advice. The data set used in this paper are the bearing degradation data collected by the Intelligent Maintenance Center of the University of Cincinnati with Every 10 minutes. In order to make the prediction time consistent with the actual time, so this paper uses the 10min units.

Comments.5) The authors should cite new references (2016-2019 Web of Science). At least new 25 referneces. Please show that you have new knowledge. For example about fault diagnosis/naive Bayes/bearings etc.

About fault diagnosis : Fault diagnosis of single-phase induction motor based on acoustic signals. Multi-dimensional variational mode decomposition for bearing-crack detection in wind turbines with large driving-speed variations. Optimised ensemble empirical mode decomposition with optimised noise parameters and its application to rolling element bearing fault diagnosis.

About NB: Diagnostics of Rotor Damages of Three-Phase Induction Motors Using Acoustic Signals and SMOFS-20-EXPANDED

About Weibull :Synthetic fault factor features under Weibull stochastic interference

Response: Thanks for the good suggestion! we read the literature you recommended about fault diagnosis/Naive Bayes /weibull etc. and cited it.

Reviewer 2 Report

Do mandatory changes in attached file

Author Response

Response to the referee’s comments

Thank you very much for your comments about our paper(Manuscript ID: entropy-396966. Title: Bearing remaining useful life prediction based on Naive Bayes and Weibull distributions)submitted to Entropy. We have checked the manuscript and revised it according to the comments. We submit here the revised manuscript as well as a list of changes. If you have any question about this paper, please don’t hesitate to let me know. Your efforts in the review process of this manuscript are greatly appreciated.

Comments.1) Several techniques dealt with a priori or a posteriori reliability updating. The origin was the keywork: DOI: 10.1016/0266-8920(95)00030-5, applied to one of a kind systems. Update reliabilitycan be performed by modeling of by monitoring. Bearing monitoring is a classic topic, your ideas are good, but the state of the art and explanations need to be near-to-perfect.

Response: Thanks for the good suggestion! We have added this question to the paper(Refer to lines 39 to 45 in the paper).  

.

Comments.2) Your work must accomplish a better state of the art, including this historical hint along with: Do you have some experience about slewing bearing? This is a rotational rolling-element bearing that typically supports a heavy but slow-turning or slow-oscillating load, often a horizontal platform such as a conventional crane, a swing yarder, or the wind-facing platform of a horizontal-axis windmill.   

Response: Thank you for your question. we have no experience about slewing bearing. The experimental platform used in this paper is four double row bearings (Bearing 1-4, model: ZA-2115) on the shaft and all the bearings were lubricated with an oil circulation system that regulated the flow and the temperature of the lubricant. A magnetic plug installed in the oil feedback pipe was used to collect debris from the oil as evidence of bearing degradation. When the accumulated debris adhered to the magnetic plug exceeded a certain level, an electrical switch turned off and the test stopped. 

Comments.3) Use of AI techniques, check and include the use of Journal of manufacturing systems 48, 108-121, because the idea is very new or ANN in mechanical applications, such as The International Journal of Advanced Manufacturing Technology 83 (5-8), 847-859 Some people in Mexico are working about how to model using different approaches, in second order equation systems. You aim your work at experimental monitoring, anyway models can be cited at least a little.

Response: Thanks for your comments. We have added this question to the paper and cited in the paper.

Comments.4) FIGURE 2: INCLUDE MORE LABELS, TO EXPLAIN BETTER.

.Response: Thanks for your time and effort. We have added more descriptions of figure 2.

Comments.5) Conclusions are too short; please define them as bullets, one per each highlight.

Response: Thanks for the good suggestion! We have added conclusions in the paper.

Round 2

Reviewer 1 Report

-Fig 4-6 please add labels to axes OX and OY for exaple: number of feature, value of feature

Author Response

Response to the referee’s comments

Thank you very much for your comments about our paper(Manuscript ID: entropy-396966. Title: Bearing remaining useful life prediction based on Naive Bayes and Weibull distributions)submitted to Entropy. We have checked the manuscript and revised it according to the comments. We submit here the revised manuscript as well as a list of changes. If you have any question about this paper, please don’t hesitate to let me know. Your efforts in the review process of this manuscript are greatly appreciated.

Comments.1) Fig 4-6 please add labels to axes OX and OY for example: number of feature, value of feature.

Response: Thank you for your question. We have revised the labels to axes OX and OY of Figure 4-6 in our paper.

Reviewer 2 Report

Please, don´t do again the same mistake: bearings were studied along the last 50 years (even more, AFMBA was founded in the 50s). Please do all previous changes, include the cites and make a correlation of modles with experiemntal tests.

Author Response

Response to the referee’s comments

Thank you very much for your comments about our paper(Manuscript ID: entropy-396966. Title: Bearing remaining useful life prediction based on Naive Bayes and Weibull distributions)submitted to Entropy. We have checked the manuscript and revised it according to the comments. We submit here the revised manuscript as well as a list of changes. If you have any question about this paper, please don’t hesitate to let me know. Your efforts in the review process of this manuscript are greatly appreciated.

Comments.1) Several techniques dealt with a priori or a posteriori reliability updating. The origin was the keywork: DOI: 10.1016/0266-8920(95)00030-5, applied to one of a kind systems. Update reliability can be performed by modeling of by monitoring. Bearing monitoring is a classic topic, your ideas are good, but the state of the art and explanations need to be near-to-perfect.

Response: Thanks for the good suggestion! The origin work(DOI:10.1016/0266-8920(95)00030-5)is model-based method to measure the life prediction of the system. The method is in our paper is mainly based on the data-driven method for reliability verification. The work (DOI:10.1016/0266-8920(95)00030-5) is well, so we cite it in our paper(Refer to lines 41 to 43 in the text).

Comments.2) Your work must accomplish a better state of the art, including this historical hint along with: Do you have some experience about slewing bearing? This is a rotational rolling-element bearing that typically supports a heavy but slow-turning or slow-oscillating load, often a horizontal platform such as a conventional crane, a swing yarder, or the wind-facing platform of a horizontal-axis windmill.  

Response: Thank you for your question. Slewing bearing is the experimental platform that you describe. The bearing test experimental platform used in this paper is mainly provided by the Intelligent Maintenance Center of the University of Cincinnati (Figure 2) and its description (Refer to lines 144 to 155 in the text). This paper mainly analyzed the full cycle data of the Intelligent Maintenance Center of the University of Cincinnati, and through the analysis of the data, predicted the rest life of the bearing.

Comments.3) Use of AI techniques, check and include the use of Journal of manufacturing systems 48, 108-121, because the idea is very new or ANN in mechanical applications, such as The International Journal of Advanced Manufacturing Technology 83 (5-8), 847-859 Some people in Mexico are working about how to model using different approaches, in second order equation systems. You aim your work at experimental monitoring, anyway models can be cited at least a little.

Response: Thanks for your comments. AI techniques and ANN is a popular and novel method in predicting the residual life of machinery. Therefore, the new residual life prediction method is analyzed in the paper [Refer to lines 90 to 95 in the text].

Comments.4) FIGURE 2: INCLUDE MORE LABELS, TO EXPLAIN BETTER.

.Response: Thanks for your time and effort. We have added more descriptions of figure 2(Refer to lines 141 to 152 in the text).

Comments.5) Conclusions are too short; please define them as bullets, one per each highlight.

Response: Thanks for the good suggestion! We have added conclusions in the paper.

Comments.6) The a priori and a posterior reliability approaches are not included. Works from the 90s key in this ideas were not included.

Response: Thanks for your time and effort. In this article, the a priori and a posterior reliability approaches is introduced in Chapter 2.3 (Naive Bayes). Prior knowledge is mainly determined by formula (6). Posterior knowledge is determined by formula (10). The method of this paper is based on the prior knowledge of the known bearing data, and the training model of the prior knowledge of the bearing is obtained. When the sample data set is added, the reliability is verified according to the property of the Naive Bayes maximum posterior knowledge.

Comments.7) Modelling: how to check models with experimental campaign is the key: Chebyshev polinomials, homotophy, and other methods were developed in the recent past. Just now is another experimental approach. Authors must consider ideas and references gave in previous comments. State of the art does not include previous suggestions: the updating of models by means of experimental measurements are key today. Bearings are very studied along history: from the eraly 50s. the only way to make paper better is to follow and do all previous changes.

Response: Thanks for the good suggestion! Bearing research has a long history. The research of bearing reliability is mainly divided into model-based and data-based methods. Cox put forward the use of failure rate model for reliability prediction in 1972. Subsequently, many scholars use model-based methods to predict the residual life of bearings, as shown in the paper [Refer to lines 38 to 51 in the text]. However, as the complexity of the system increases, the construction of failure models becomes more and more difficult. The data-driven method does not need to build a complex physical model, but mainly extracts a large amount of historical data and then extracts fault information for data analysis, so as to predict the remaining life of the bearing. Due to its simple method, it has been widely concerned and used in the paper. It is analyzed in this paper. Some achievements have been made in the recent research on the residual life of bearing, see paper [Refer to lines 89 to 97 in the text]. However, the existing data-driven residual life prediction method does not accumulate knowledge to determine the bearing state. Health status determination is based on expert experience. Bayes is a datahl-driven method based on prior knowledge, which effectively avoids the randomness of results. So this paper uses a data-driven approach based on naive Bayes.

Comments.8) A complete fatigue testing campaign is demanded.

Response: Thanks for your time and effort. A complete fatigue testing campaign is demanded. In this paper, using the Intelligent Maintenance Center of the University of Cincinnati bearing equipment (Figure 2) acquisition bearing the whole cycle of data (run-to-failure). The work of this paper is to analyze the collected fatigue data set and predict the remaining life of bearing.
